# Triple Wins: Boosting Accuracy, Robustness and Efficiency Together by Enabling Input-Adaptive Inference

**Ting-Kuei Hu**,[*] **Tianlong Chen**,[*] **Haotao Wang** **Zhangyang Wang**
Department of Computer Science and Engineering
Texas A&M University, USA
`{tkhu,wiwjp619,htwang,atlaswang}@tamu.edu`

## Abstract

Deep networks were recently suggested to face the odds between accuracy (on clean natural images) and robustness (on adversarially perturbed images) (Tsipras et al., 2019). Such a dilemma is shown to be rooted in the inherently higher sample complexity (Schmidt et al., 2018) and/or model capacity (Nakkiran, 2019), for learning a high-accuracy and robust classifier. In view of that, give a classification task, growing the model capacity appears to help draw a *win-win* between accuracy and robustness, yet at the expense of model size and latency, therefore posing challenges for resource-constrained applications. Is it possible to *co-design* model accuracy, robustness and efficiency to achieve their *triple wins*?

This paper studies multi-exit networks associated with input-adaptive efficient inference, showing their strong promise in achieving a "sweet point" in co-optimizing model accuracy, robustness and efficiency. Our proposed solution, dubbed *Robust Dynamic Inference Networks* (RDI-Nets), allows for each input (either clean or adversarial) to adaptively choose one of the multiple output layers (early branches or the final one) to output its prediction. That multi-loss adaptivity adds new variations and flexibility to adversarial attacks and defenses, on which we present a systematical investigation. We show experimentally that by equipping existing backbones with such robust adaptive inference, the resulting RDI-Nets can achieve better accuracy and robustness, yet with over 30% computational savings, compared to the defended original models.

## 1 Introduction

Deep networks, despite their high predictive accuracy, are notoriously vulnerable to adversarial attacks (Goodfellow et al., 2015; Biggio et al., 2013; Szegedy et al., 2014; Papernot et al., 2016). While many defense methods have been proposed to increase a model's *robustness* to adversarial examples, they were typically observed to hamper its *accuracy* on original clean images. Tsipras et al. (2019) first pointed out the inherent tension between the goals of adversarial robustness and standard accuracy in deep networks, whose provable existence was shown in a simplified setting. Zhang et al. (2019) theoretically quantified the accuracy-robustness trade-off, in terms of the gap between the risk for adversarial examples versus the risk for non-adversarial examples.

It is intriguing to consider whether and why the model accuracy and robustness have to be at odds. Schmidt et al. (2018) demonstrated that the number of samples needed to achieve adversarially robust generalization is polynomially larger than that needed for standard generalization, under the adversarial training setting. A similar conclusion was concurred by Sun et al. (2019) in the standard training setting. Tsipras et al. (2019) considered the accuracy-robustness trade-off as an inherent trait of the data distribution itself, indicating that this phenomenon persists even in the limit of infinite data. Nakkiran (2019) argued from a different perspective, that the complexity (e.g. capacity) of a robust classifier must be higher than that of a standard classifier. Therefore, replacing a larger-capacity classifier might effectively alleviate the trade-off. Overall, those existing works appear to suggest that, while accuracy and robustness are likely to trade off for a fixed classification model

---

[*]Equal contribution

and on a given dataset, such trade-off might be effectively alleviated ("win-win"), if supplying more training data and/or replacing a larger-capacity classifier.

On a separate note, deep networks also face the pressing challenge to be deployed on resource-constrained platforms due to the prosperity of smart Internet-of-Things (IoT) devices. Many IoT applications naturally demand security and trustworthiness, e.g., , biometrics and identity verification, but can only afford limited latency, memory and energy budget. Hereby we extend the question: *can we achieve a triple-win, i.e., , an accurate and robust classfier while keeping it efficient?*

This paper makes an attempt in providing a positive answer to the above question. Rather than proposing a specific design of robust light-weight models, we reduce the average computation loads by input-adaptive routing to achieve triple-win. To this end, we introduce the input-adaptive dynamic inference (Teerapittayanon et al., 2017; Wang et al., 2018a), an emerging efficient inference scheme in contrast to the (non-adaptive) model compression, to the adversarial defense field for the first time. Given any deep network backbone (e.g., , ResNet, MobileNet), we first follow (Teerapittayanon et al., 2017) to augment it with multiple early-branch output layers in addition to the original final output. Each input, regardless of clean or adversarial samples, adaptively chooses which output layer to take for its own prediction. Therefore, a large portion of input inferences can be terminated early when the samples can already be inferred with high confidence.

Up to our best knowledge, no existing work studied adversarial attacks and defenses for an adaptive multi-output model, as the multiple sources of losses provide much larger flexibility to compose attacks (and therefore defenses), compared to the typical single-loss backbone. We present a systematical exploration on how to (white-box) attack and defense our proposed multi-output network with adaptive inference, demonstrating that the composition of multiple-loss information is critical in making the attack/defense strong. Fig. 1 illustrates our proposed *Robust Dynamic Inference Networks* (RDI-Nets). We show experimentally that the input-adaptive inference and multi-loss flexibility can be our friend in achieving the desired "triple wins". With our best defended RDI-Nets, we achieve better accuracy and robustness, yet with over 30% inference computational savings, compared to the defended original models as well as existing solutions co-designing robustness and efficiency (Gui et al., 2019; Guo et al., 2018). The codes can be referenced from *https://github.com/TAMU-VITA/triple-wins.*

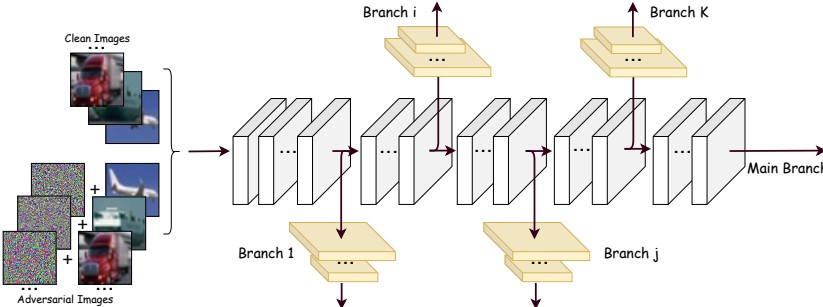

Figure 1: Our proposed RDI-Net framework, a **defended** multi-output network enabling dynamic inference. Each image, being it clean or adversarially perturbed, adaptively picks one branch to exit.

## 2 RELATED WORK

### 2.1 ADVERSARIAL DEFENSE

A magnitude of defend approaches have been proposed (Kurakin et al., 2017; Xu et al., 2018; Song et al., 2018; Liao et al., 2018), although many were quickly evaded by new attacks (Carlini & Wagner, 2017; Baluja & Fischer, 2018). One strong defense algorithm that has so far not been fully compromised is adversarial training (Madry et al., 2018). It searches for adversarial images to augment the training procedure, although at the price of higher training costs (but not affecting inference efficiency). However, almost all existing attacks and defenses focus on a single-output classification (or other task) model. We are unaware of prior studies directly addressing attacks/defenses to more complicated networks with multiple possible outputs.

One related row of works are to exploit model ensemble (Tramèr et al., 2018; Strauss et al., 2017) in adversarial training. The gains of the defended ensemble compared to a single model could be viewed as the benefits of either the benefits of diversity (generating stronger and more transferable perturbations), or the increasing model capacity (consider the ensembled multiple models as a compound one). Unfortunately, ensemble methods could amplify the inference complexity and be detrimental for efficiency. Besides, it is also known that injecting randomization at inference time helps mitigate adversarial effects (Xie et al., 2018; Cohen et al., 2019). Yet up to our best knowledge, no work has studied non-random, but rather input-dependent inference for defense.

## 2.2 Efficient Inference

Research in improving deep network efficiency could be categorized into two streams: the *static* way that designs compact models or compresses heavy models, while the compact/compressed models remain fixed for all inputs at inference; and the *dynamic* way, that at inference the inputs can choose different computational paths adaptively, and the simpler inputs usually take less computation to make predictions. We briefly review the literature below.

*Static: Compact Network Design and Model Compression.* Many compact architectures have been specifically designed for resource-constrained applications, by adopting lightweight depthwise convolutions (Sandler et al., 2018), and group-wise convolutions with channel-shuffling (Zhang et al., 2018), to just name a few. For model compression, Han et al. (2015) first proposed to sparsify deep models by removing non-significant synapses and then re-training to restore performance. Structured pruning was later on introduced for more hardware friendliness (Wen et al., 2016). Layer factorization (Tai et al., 2016; Yu et al., 2017), quantization (Wu et al., 2016), model distillation (Wang et al., 2018c) and weight sharing (Wu et al., 2018) have also been respectively found effective.

*Dynamic: Input-Adaptive Inference.* Higher inference efficiency could be also accomplished by enabling input-conditional execution. Teerapittayanon et al. (2017); Huang et al. (2018); Kaya et al. (2019) leveraged intermediate features to augment multiple side branch classifiers to enable early predictions. Their methodology sets up the foundation for our work. Other efforts (Figurnov et al., 2017; Wang et al., 2018a;b; 2019) allow for an input to choose between passing through or skipping each layer. The approach could be integrated with RDI-Nets too, which we leave as future work.

## 2.3 Bridging Robustness with Efficiency

A few studies recently try to link deep learning robustness and efficiency. Guo et al. (2018) observed that in a sparse deep network, appropriately sparsified weights improve robustness, whereas over-sparsification (e.g., less than 5% nonzero weights) in turn makes the model more fragile. Two latest works (Ye et al., 2019; Gui et al., 2019) examined the robustness of compressed models, and concluded similar observations that the relationship between mode size and robustness depends on compression methods and are often non-monotonic. Lin et al. (2019) found that activation quantization may hurt robustness, but can be turned into effective defense if enforcing continuity constraints.

Different from above methods that tackle robustness from *static* compact/compressed models, the proposed RDI-Nets are the first to address robustness from the *dynamic* input-adaptive inference. Our experiment results demonstrate the consistent superiority of RDI-Nets over those static methods (Section 4.3). Moreover, applying *dynamic* inference top of those static methods may further boost the robustness and efficiency, which we leave as future work.

## 3 Approach

With the goal of achieving inference efficiency, we first look at the setting of multi-output networks and the specific design of RDI-Net in Section 3.1. Then we define three forms of adversarial attacks for multi-output networks in Section 3.2 and their corresponding defense methods in Section 3.3.

Note that RDI-Nets achieve "triple wins"via reducing the average computation loads through input-adaptive routing. It is not to be confused with any specifically-designed robust light-weight model.

## 3.1 Designing RDI-Nets for Higher Inference Efficiency

Given an input image $x$, an $N$-output network can produce a set of predictions $[\hat{y_1}, ..., \hat{y_N}]$ by a set of transformations $[f_{\theta_1}(\cdot), ..., f_{\theta_N}(\cdot)]$. $\theta_i$ denote the model parameter of $f_{\theta_i}$, $i = 1, ..., N$, and $f_{\theta_i}$s

will typically share some weights. With an input $x$, one can express $\hat{y}_i = f_{\theta_i}(x)$. We assume that the final prediction will be one **chosen** (NOT fused) from $[\hat{y}_1, ..., \hat{y}_N]$ via some deterministic strategy.

We now look at RDI-Nets as a specific instance of multi-output networks, specifically designed for the goal of more efficient, input-adaptive inference. As shown in Fig. 1, **for any deep network** (e.g., , ResNet, MobileNet), we could append $K$ side branches (with negligible overhead) to allow for early-exit predictions. In other words, it becomes a $(K+1)$-output network, and the *subneworks* with the $K+1$ exits, from the lowest to the highest (the original final output), correspond to $[f_{\theta_1}(\cdot), ..., f_{\theta_{K+1}}(\cdot)]$. They share their weights in a *nested* fashion: $\theta_1 \subseteq \theta_2 ... \subseteq \theta_{K+1}$, with $\theta_{K+1}$ including the entire network's parameters.

Our deterministic strategy in selecting one final output follows (Teerapittayanon et al., 2017). We set a confidence threshold $t_k$ for each $k$-th exit, $k = 1, ..., K + 1$, and each input $x$ will terminate inference and output its prediction in the earliest exit (smallest $k$), whose softmax entropy (as a confidence measure) falls below $t_k$. All computations after the $k$-th exit will not be activated for this $x$. Such a progressive and early-halting mechanism effectively saves unnecessary computation for most easier-to-classify samples, and applies in both training and inference. Note that, if efficiency is not the concern, instead of choosing (the earliest one), we could have designed an adaptive or randomized fusion of all $f_{\theta_i}$ predictions: but that falls beyond the goal of this work.

The training objective for RDI-Nets could be written as

$$L_{RDI} = \sum_{i=1}^{K+1} w_i [(\phi(f_i(\theta_i|x), y) + \phi(f_i(\theta_i|x^{adv}), y)], \tag{1}$$

For each exit loss, we minimize a hybrid loss of accuracy (on clean $x$) and robustness (on $x^{adv}$). The $K + 1$ exits are balanced with a group of weights $\{w_i\}_{i=1}^{K+1}$. More details about RDI-Net structures, hyperparameters, and inference branch selection can be founded in Appendix A, B, and C.

In what follows, we discuss three ways to generate $x^{adv}$ in RDI-Nets, and then their defenses.

## 3.2 THREE ATTACK FORMS ON MULTI-OUTPUT NETWORKS

We consider white box attacks in this paper. Attackers have access to the model's parameters, and aim to generate an adversarial image $x^{adv}$ to fool the model by perturbing an input $x$ within a given magnitude bound.

We next discuss three *attack forms* for an $N$-output network. Note that they are independent of, and to be distinguished from *attacker algorithms* (e.g., , PGD, C&W, FGSM): the former depicts the *optimization formulation*, that can be solved any of the attacker algorithms.

**Single Attack**    Naively extending from attacking single-output networks, a single attack is defined to maximally fool one $f_{\theta_i}(\cdot)$ only, expressed as:

$$x_i^{adv} = \underset{x' \in |x' - x|_\infty \leq \epsilon}{\arg\max} |\phi(f_{\theta_i}(x'), y)|, \tag{2}$$

where $y$ is the ground truth label, and $\phi$ is the loss for $f_{\theta_i}$ (we assume softmax for all). $\epsilon$ is the perturbation radius and we adopt $\ell_\infty$ ball for an empirically strong attacker. Naturally, an $N$-output network can have $N$ different single attacks. However, each single attack is derived without being aware of other parallel outputs. The found $x_i^{adv}$ is not necessarily transferable to other $f_{\theta_j}$s ($j \neq i$), and therefore can be easily bypassed if $x$ is re-routed through other outputs to make its prediction.

**Average Attack**    Our second attack maximizes the average of all $f_{\theta_i}$ losses, so that the found $x^{adv}$ remains in effect no matter which one $f_{\theta_i}$ is chosen to output the prediction for $x$:

$$x_{avg}^{adv} = \underset{x' \in |x' - x|_\infty \leq \epsilon}{\arg\max} |\frac{1}{N} \sum_{j=1}^{N} \phi(f_{\theta_j}(x'), y)|, \tag{3}$$

The average attack addresses takes into account the attack transferablity and involves all $\theta_j$s into optimization. However, while only one output will be selected for each sample at inference, the average strategy might weaken the individual defense strength of each $f_{\theta_i}$.

**Max-Average Attack**   Our third attack aims to emphasize individual output defense strength, more than simply maximizing an all-averaged loss. We first solve the $N$ single attacks $x_i^{adv}$ as described in Eqn. 2, and denote their collection as $\Omega$. We then solve the max-average attack via the following:

$$x_{max}^{adv} \leftarrow x_{i^*}^{adv}, \text{ where } x_{i^*}^{adv} \in \Omega \text{ and } i^* = \arg\max_i |\frac{1}{N} \sum_{j=1}^{N} \phi(f_{\theta_j}(x_i^{adv}), y)|. \qquad (4)$$

Note Eqn. 4 differs from Eqn. 3 by adding an $\Omega$ constraint to balance between "commodity" and "specificity". The found $x_{max}^{adv}$ **both** strongly increases the averaged loss values from all $f_i$s (therefore possessing transferablity), **and** maximally fools one individual $f_{\theta_i}$s as it is selected from the collection $\Omega$ of single attacks.

## 3.3   DEFENCE ON MULTI-OUTPUT NETWORKS

For simplicity and fair comparison, we focus on adversarial training (Madry et al., 2018) as our defense framework, where the three above defined attack forms can be plugged-in to generate adversarial images to augment training, as follows ($\Theta$ is the union of learnable parameters):

$$\theta_i \in \Theta, \text{ where } \theta_i = \arg\min_{\theta'} |\phi(f_{\theta_i}(x), y) + \phi(f_{\theta_i}(x^{adv}), y)|., \qquad (5)$$

where $x^{adv} \in \{x_i^{adv}, x_{avg}^{adv}, x_{max}^{adv}\}$. As $f_i$s partially share their weights $\theta_i$ in a multi-output network, the updates from different $f_i$s will be averaged on the shared parameters.

## 4   EXPERIMENTAL RESULTS

### 4.1   EXPERIMENTAL SETUP

**Evaluation Metrics**   We evaluate accuracy, robustness, and efficiency, using the metrics below:
- Testing Accuracy (**TA**): the classification accuracy on the original clean test set.
- Adversarial Testing Accuracy (**ATA**): Given an attacker, ATA stands for the classification accuracy on the attacked test set. It is the same as the "robust accuracy" in (Zhang et al., 2019).
- Mega Flops (**MFlops**): The number of million floating-point multiplication operations consumed on the inference, averaged over the entire testing set.

**Datasets and Benchmark Models**   We evaluate three representative CNN models on two popular datasets: SmallCNN on MNIST (Chen et al., 2018); ResNet-38 (He et al., 2016) and MobileNet-V2 (Sandler et al., 2018) on CIFAR-10. The three networks span from simplest to more complicated, and covers a compact backbone. All three models are defended by adversarial training, constituting **strong baselines**. Table 1 reports the models, datasets, the attacker algorithm used in attack & defense, and thee TA/ATA/MFlops performance of three defended models.

**Attack and Defense on RDI-Nets**   We build RDI-Nets by appending side branch outputs for each backbone. For SmallCNN, we add two side branches ($K = 2$). For ResNet-38 and MobileNet-V2, we have $K = 6$ and $K = 2$, respectively. The branches are designed to cause negligible overheads: more details of their structure and positions can be referenced in Appendix B. We call those result models RDI-SmallCNN, RDI-ResNet38 and RDI-MobileNetV2 hereinafter.

We then generate attacks using our three defined forms. Each attack form could be solved with various attacker algorithms (e.g., PGD, C&W, FGSM), and by default we solve it with the same attacker used for each backbone in Table 1. If we fix one attacker algorithm (e.g., PGD), then TA/ATA for a single-output network can be measured without ambiguity. Yet for $(K+1)$-output RDI-Nets, there could be at least $K+3$ different ATA numbers for one defended model, depending on what attack form in Section 3.1 to apply ($K+1$ single attacks, 1 average attack, and 1 max-average attack). For example, we denote by **ATA (Branch1)** the ATA number when applying the single attack generated from the first side output branch (e.g., $x_1^{adv}$); similarly elsewhere.

We also defend RDI-Nets using adversarial training, using the forms of adversarial images to augment training. By default, we adopt three adversarial training defense schemes: **Main Branch** (single attack using $x_{K+1}^{adv}$)[1], **Average** (using $x_{avg}^{adv}$), and **Max-Average** (using $x_{avg}^{max}$), in addition to the undefended RDI-Nets (using standard training) denoted as **Standard**.

---

[1] We tried adversarial training using other $K$ earlier side branch single attacks, and found their TA/ATA to be much more deteriorated compared to the main branch one. We thus report this only for compactness.

We cross evaluate ATAs of different defenses and attacks, since an ideal defense shall protect against all possible attack forms. To faithfully indicate the actual robustness, we choose the *lowest number* among all $K + 3$ ATAs, denoted as **ATA (Worst-Case)**, as the robustness measure for an RDI-Net.

Table 1: Benchmarking results of adverserial training of three networks. PGD-40 denotes running the projected gradient descent attacker (Madry et al., 2018) for 40 iterations. We set the perturbation size as $0.3$ for MNIST and $8/255$ for CIFAR-10 in $\ell_\infty$ norm (adopted by all following experiments).

| Model | Dataset | Defend | Attack | TA | ATA | MFlops |
|---|---|---|---|---|---|---|
| SmallCNN | MNIST | PGD-40 | PGD-40 | 99.49% | 96.31% | 9.25 |
| ResNet-38 | CIFAR-10 | PGD-10 | PGD-20 | 83.62% | 42.29% | 79.42 |
| MobileNetV2 | CIFAR-10 | PGD-10 | PGD-20 | 84.42% | 46.92% | 86.91 |

## 4.2 EVALUATION AND ANALYSIS

**MNIST Experiments** The MNIST experimental results on RDI-SmallCNN are summarized in table 2, with several meaningful observations to be drawn. First, the undefended models (Standard) are easily compromised by all attack forms. Second, The single attack-defended model (Main Branch) achieves the best ATA against the same type of attack, i.e., ATA (Main Branch), and also seems to boost the closest output branch's robustness, i.e., ATA (Branch 2). However, its defense effect on the further-away Branch 1 is degraded, and also shows to be fragile under two stronger attacks (Average, and Max-Average). Third, both Average and Max-Average defenses achieve good TAs, as well as ATAs against all attack forms (and therefore Worst-Case), with Max-Average slightly better at both (the margins are small due to the data/task simplicity; see next two).

Moreover, compared to the strong baseline of SmallCNN defended by PGD (40 iterations)-based adversarial training, RDI-SmallCNN with Max-Average defense wins in terms of both TA and ATA. Impressively, that comes together with 34.30% computational savings compared to the baseline. Here the different defense forms do not appear to alter the inference efficiency much: they all save around 34% - 36% MFlops compared to the backbone.

Table 2: The performance of RDI-SmallCNN. The "**Average MFlops**" is calculated by averaging the total flop costs consumed over the inference of the entire set (different samples take different FLOPs due to input-adaptive inference). The perturbation size and step size are $0.3$ and $0.01$, respectively.

| Defense Method | Standard | Main Branch | Average | Max-Average |
|---|---|---|---|---|
| **TA** | 99.48% | 99.50% | 99.51% | **99.52%** |
| ATA (Branch 1) | 6.60% | 60.50% | 98.69% | 98.52% |
| ATA (Branch 2) | 3.16% | 98.14% | 97.64% | 97.62% |
| ATA (Main Branch) | 1.32% | 96.70% | 96.30% | 96.43% |
| ATA (Average) | 2.61% | 61.35% | 97.37% | 97.42% |
| ATA (Max-Average) | 2.10% | 61.83% | 96.82% | 96.89% |
| **ATA (Worst-Case)** | 1.32% | 60.50% | 96.30% | **96.43%** |
| Average MFlops | 5.89 | 5.89 | 5.95 | 6.08 |
| **Computation Saving** | 36.40% | 36.40% | 35.70% | 34.30% |

**CIFAR-10 Experiments** The results on RDI-ResNet38 and RDI-MobileNetV2 are presented in Tables 3 and 4, respectively. Most findings seem to concur with MNIST experiments. Specifically, on the more complicated CIFAR-10 classification task, Max-Average defense achieves much more obvious margins over Average defense, in terms of ATA (Worst-Case): 2.79% for RDI-ResNet38, and 1.06% for RDI-MobileNetV2. Interestingly, the Average defense is not even the strongest in defending average attacks, as Max-Average defense can achieve higher ATA (Average) in both cases. We conjecture that averaging all branch losses might "over-smooth" and diminish useful gradients.

Compared to the defended ResNet-38 and MobileNet-V2 backbones, RDI-Nets with Max-Average defense achieve higher TAs and ATAs for both. Especially, the ATA (Worst-Case) of RDI-ResNet-38 surpasses the ATA of ResNet-38 defended by PGD-adversarial training by **1.03%**, while saving around **30%** inference budget. We find that different defenses on CIFAR-10 have more notable impacts on computational saving. Seemingly, a stronger defense (Max-Average) requires inputs to go through the scrutiny of more layers on average, before outputting confident enough predictions: a sensible observation as we expect.

**Visualization of Adaptive Inference Behaviors** We visualize the exiting behaviors of RDI-ResNet38 in Fig 4.2. We plot each branch exiting percentage on clean set and adversarial sets (worst-case) of examples. A few interesting observations can be found. First, we observe that the single-attack defended model can be easily fooled as adversarial examples can be routed through other less-defended outputs (due to the limited transferability of attacks between different outputs). Second, the two stronger defenses (Average and Max-Average) show much more uniform usage of multiple outputs. Their routing behaviors for clean examples are almost identical. For adversarial examples, Max-Average tends to call upon the full inference more often (i.e., more "conservative").

Table 3: The performance evaluation on RDI-ResNet38. The perturbation size and step size are $8/255$ and $2/255$, respectively.

| Defence Method | Standard | Main Branch | Average | Max-Average |
|---|---|---|---|---|
| **TA** | 92.43% | 83.74% | 82.42% | **83.79%** |
| ATA (Branch1) | 0.12% | 12.02% | 71.56% | 69.71% |
| ATA (Branch2) | 0.01% | 5.58% | 66.67% | 63.11% |
| ATA (Branch3) | 0.04% | 42.73% | 60.65% | 60.72% |
| ATA (Branch4) | 0.06% | 34.95% | 50.17% | 47.82% |
| ATA (Branch5) | 0.06% | 41.77% | 44.83% | 45.53% |
| ATA (Branch6) | 0.11% | 41.68% | 45.83% | 44.12% |
| ATA (Main Branch) | 0.13% | 42.74% | 47.52% | 49.82% |
| ATA (Average) | 0.01% | 9.14% | 42.09% | 43.32% |
| ATA (Max-Average) | 0.01% | 7.15% | 40.53% | 43.43% |
| **ATA (Worst-Case)** | 0.01% | 5.58% | 40.53% | **43.32%** |
| Average MFlops | 29.41 | 48.27 | 56.90 | 57.81 |
| **Computation Saving** | 62.96% | 39.20% | 28.35% | 27.20% |

Table 4: The performance evaluation on RDI-MobilenetV2. The perturbation size and step size are $8/255$ and $2/255$, respectively.

| Defence Method | Standard | Main Branch | Average | Max-Average |
|---|---|---|---|---|
| **TA** | 93.22% | **85.28%** | 82.14% | 84.91% |
| ATA (Branch1) | 0.35% | 37.40% | 67.65% | 71.78% |
| ATA (Branch2) | 0% | 47.35% | 50.38% | 50.15% |
| ATA (Main Branch) | 0% | 46.69% | 49.33% | 46.99% |
| ATA (Average) | 0% | 35.20% | 45.93% | 47.00% |
| ATA (Max-Average) | 0% | 36.66% | 49.33% | 50.18% |
| **ATA (Worst-Case)** | 0% | 35.20% | 45.93% | **46.99%** |
| Average MFlops | 49.78 | 52.81 | 58.23 | 60.84 |
| **Computation Saving** | 42.72% | 39.23% | 33.00% | 29.99% |

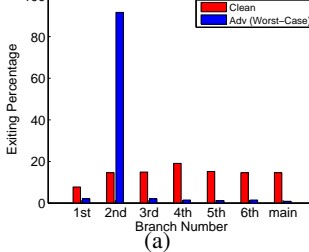 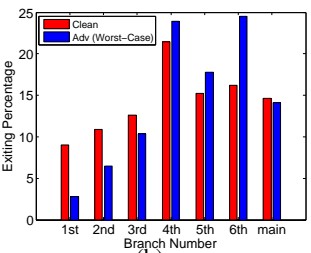 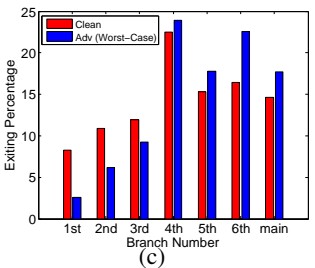

Figure 2: The exiting behaviours of RDI-ResNet38 defended by (a) Single attack defense (Main Branch); (b) Average defense; and (c) Max-Average defense.

## 4.3 COMPARISON WITH DEFENDED SPARSE NETWORKS

An alternative to achieve accuracy-robust-efficiency trade-off is by defending a sparse or compressed model. Inspired by (Guo et al., 2018; Gui et al., 2019), we compare RDI-Net with Max-Average

defense to the following baseline: first compressing the network with a state-of-the-art model compression method (Huang & Wang, 2018), and then defend the compressed network using the PGD-10 adversarial training. We sample different sparsity ratios in (Huang & Wang, 2018) to obtain models of different complexities. Fig. 6 in Appendix visualizes the comparison on ResNet-38: for either method, we sample a few models of different MFLOPs. At similar inference costs (e.g., 49.38M for pruning + defense, and 48.35M for RDI-Nets), our proposed approach consistently achieves higher ATAs (> 2%) than the strong pruning + defense baseline, with higher TAs.

We also compare with the latest ATMC algorithm (Gui et al., 2019) that jointly optimizes robustness and efficiency, applied the same ResNet-38 backbone. As shown in Table 5, at comparable MFlops, RDI-ResNet-38 surpasses ATMC by 0.3% in terms of ATA, with a similar TA.

Table 5: Performance comparison between RDI-ResNet38 and ATMC.

| Methods | TA | ATA | MFlops |
|---|---|---|---|
| ATMC (Gui et al. (2019)) | **83.81** | 43.02 | 56.82 |
| RDI-ResNet-38 (**Worst-Case**) | 83.79 | **43.32** | 57.81 |

### 4.4 GENERALIZED ROBUSTNESS AGAINST OTHER ATTACKERS

In the aforementioned experiments, we have only evaluated on RDI-Nets against "deterministic" PGD-based adversarial images. We show that RDI-Nets also achieve better generalized robustness against other "randomized" or unseen attackers. We create the new "random attack": that attack will randomly combine the multi-exit losses, and summarize the results in Table 6. We also follow the similar setting in Gui et al. (2019) and report the results against FGSM (Goodfellow et al., 2015) and WRM (Sinha et al., 2018) attacker, in Tables 7 and 8 respectively (more complete results can be found in Appendix D).

Table 6: Performance on RDI-ResNet38 against random attack. The perturbation size and step size are $8/255$ and $2/255$, respectively. More details of random attack can be referenced in Appendix D.

| Defence Method | Standard | Main Branch | Average | Max-Average |
|---|---|---|---|---|
| **TA** | 92.43% | 83.74% | 82.42% | **83.79%** |
| ATA (Random) | 0.01% | 10.33% | 43.11% | **44.86%** |
| Average MFlops | 27.33 | 52.36 | 55.21 | 56.54 |
| **Computation Saving** | 65.58% | 34.07% | 30.48% | 28.80% |

Table 7: Performance on RDI-ResNet38 (defended with PGD) against FGSM attack (perturbation size is $8/255$). The original defended ResNet38 by PGD under the same attack has ATA $51.11\%$.

| Defence Method | Standard | Main Branch | Average | Max-Average |
|---|---|---|---|---|
| **TA** | 92.43% | 83.74% | 82.42% | **83.79%** |
| ATA (Main Branch) | 11.51% | 51.45% | 53.64% | 54.72% |
| ATA (Average) | 11.41% | 50.21% | 51.81% | 53.20% |
| ATA (Max-Average) | 2.09% | 47.53% | 50.63% | 52.40% |
| **ATA (Worst-Case)** | 2.09% | 47.53% | 50.63% | **51.05%** |
| Average MFlops | 65.74 | 55.27 | 58.27 | 59.67 |
| **Computation Saving** | 17.21% | 30.40% | 26.40% | 24.86% |

Table 8: Performance on RDI-ResNet38 (defended with PGD) against WRM attack (perturbation size is 0.3). The original defended ResNet38 by PGD under the same attack has ATA $83.35\%$.

| Defence Method | Standard | Main Branch | Average | Max-Average |
|---|---|---|---|---|
| **TA** | 92.43% | 83.74% | 82.42% | **83.79%** |
| ATA (Main Branch) | 34.42% | 83.74% | 82.42% | 83.78% |
| ATA (Average) | 26.48% | 83.69% | 82.36% | 83.77% |
| ATA (Max-Average) | 23.51% | 83.73% | 82.40% | 83.78% |
| **ATA (Worst-Case)** | 23.51% | 83.69% | 82.36% | **83.77%** |
| Average MFlops | 50.05 | 50.46 | 52.89 | 52.38 |
| **Computation Saving** | 36.98% | 36.46% | 33.40% | 34.04% |

## 5 DISCUSSION AND ANALYSIS

**Intuition: Multi-Output Networks as Special Ensembles**  Our intuition on defending multi-output networks arises from the success of ensemble defense in improving both accuracy and robustness (Tramèr et al., 2018; Strauss et al., 2017), which also aligns with the model capacity hypothesis (Nakkiran, 2019). A general multi-output network (Xu et al., 2019) could be decomposed by an ensemble of single-output models, with weight re-using enforced among them. It is thus more compact than an ensemble of independent models, and the extent of sharing weight calibrates ensemble diversity versus efficiency. Therefore, we expect a defended multi-output network to (mostly) inherit the strong accuracy/robustness of ensemble defense, while keeping the inference cost lower.

**Do "Triple Wins" Go Against the Model Capacity Needs?**  We point out that our seemingly "free" efficiency gains (e.g., not sacrificing TA/ATA) do not go against the current belief that a more accurate and robust classifier relies on a larger model capacity (Nakkiran, 2019). From the visualization, there remains to be a portion of clean/adversarial examples that have to utilize the full inference to predict well. In other words, the full model capacity is still necessary to achieve our current TAs/ATAs. Meanwhile, just like in standard classification (Wang et al., 2018a), not all adversarial examples are born equally. Many of them can be predicted using fewer inference costs (taking earlier exits). Therefore, RDI-Nets reduces the "effective model capacity" averaged on all testing samples for overall higher inference efficiency, while not altering the full model capacity.

## 6 CONCLUSION

This paper targets to simultaneously achieve high accuracy and robustness and meanwhile keeping inference costs lower. We introduce the multi-output network and input-adaptive dynamic inference, as a strong tool to the adversarial defense field for the first time. Our RDI-Nets achieve the "triple wins" of better accuracy, stronger robustness, and around 30% inference computational savings. Our future work will extend RDI-Nets to more dynamic inference mechanisms.

## 7 ACKNOWLEDGEMENT

We would like to thank Dr. Yang Yang from Walmart Technology for highly helpful discussions throughout this project.

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

## A  LEARNING DETAILS OF RDI-NETS

**MNIST**  We adopt the network architecture from (Chen et al., 2018) with four convolutions and three full-connected layers. We train for 13100 iterations with a batch size of 256. The learning rate is initialized as 0.033 and is lowered by 10 at 12000th and 12900th iteration. For hybrid loss, the weights $\{w_i\}_{i=1}^{N+1}$ are set as $\{1, 1, 1\}$ for simplicity. For adversarial defense/attack, we perform 40-steps PGD for both defense and evaluation. The perturbation size and step size are set as 0.3 and 0.01.

**CIFAR-10** We take ResNet-38 and MobileNetV2 as the backbone architectures. For RDI-ResNet38, we initialize learning rate as $0.1$ and decay it by a factor of 10 at 32000th and 48000th iteration. The learning procedure stops at 55000 iteration. For RDI-MobileNetV2, the learning rate is set to $0.05$ and is lowered by 10 times at 62000th and 70000th iteration. We stop the learning procedure at 76000 iteration. For hybrid loss, we follow the discussion in (Hu et al., 2019) and set $\{w_i\}_{i=1}^{N+1}$ of RDI-ResNet38 and RDI-MobileNetV2 as $\{0.5, 0.5, 0.7, 0.7, 0.9, 0.9, 2\}$ and $\{0.5, 0.5, 1\}$, respectively. For adversarial defense/attack, the perturbation size and step size are set as $8/255$ and $2/255$. 10-steps PGD is performed for defense and 20-steps PGD is utilized for evaluation.

## B  NETWORK STRUCTURE OF RDI-NETS

To build RDI-Nets, we follow the similar setting in Teerapittayanon et al. (2017) by appending additional branch classifiers at equidistant points throughout a given network, as illustrated in Fig 3, Fig 4 and Fig 5. A few pooling operations, light-weight convolutions and fully-connected layers are appended to each branch classifiers. Note that the extra flops introduced by side branch classifiers are less than 2% than the original ResNet-38 or MobileNetV2.

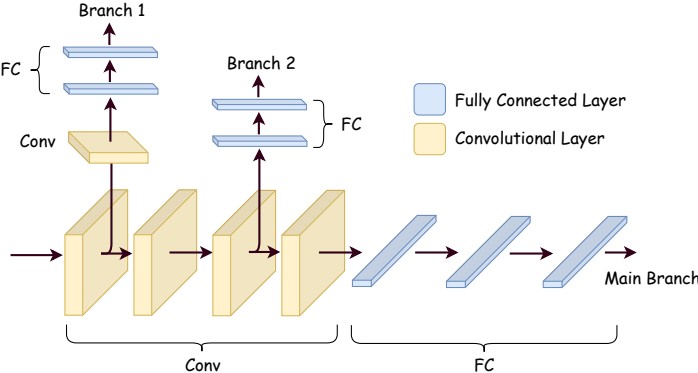

Figure 3: Network architecture of RDI-SmallCNN. Two branch classifiers are inserted after $1st$ convolutional layer and 3rd convolutional layer in the original SmallCNN.

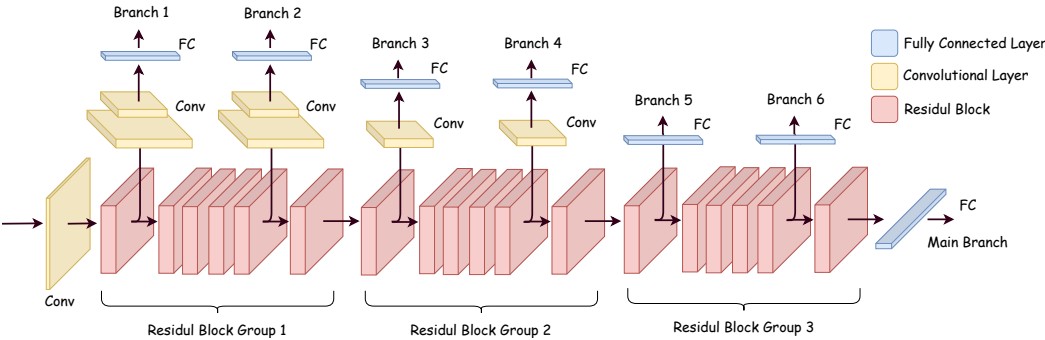

Figure 4: Network architecture of RDI-ResNet38. In each residual block group, two branch classifiers are inserted after $1st$ residual block and 4th residual block.

## C  INPUT-ADAPTIVE INFERENCE FOR RDI-NETS

Similar to the deterministic strategy in Teerapittayanon et al. (2017), we adopt the entropy as the measure of the prediction confidence. Given a prediction vector $y \in \mathbb{R}^C$, where $C$ is the number of

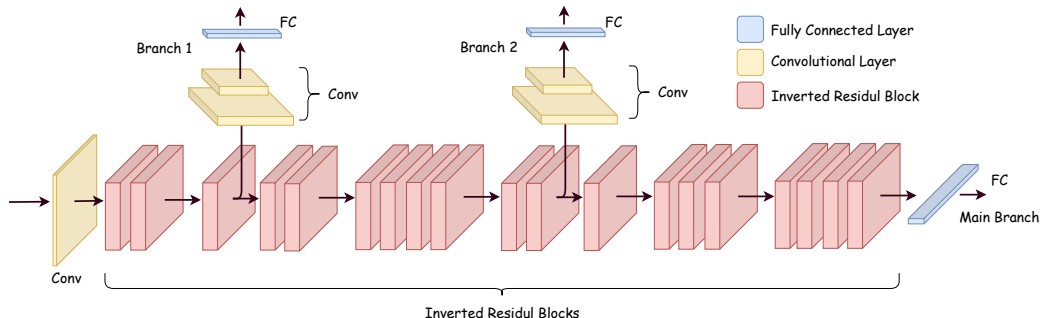

Figure 5: Network architecture of RDI-MobilenetV2. Two branch classfiers are inserted after $3rd$ inverted residual block and 11th inverted residual block in the orignal MobilenetV2.

class, the entropy of $y$ is defined as follow,

$$-\sum_{c=1}^{C}(y_c + \epsilon)log(y_c + \epsilon), \tag{6}$$

where $\epsilon$ is a small positive constant used for robust entropy computation. To perform fast inference on a $(K+1)$-output RDI-Net, we need to determine $K$ threshold numbers, i.e., $\{t_i\}_{i=1}^{K}$, so that the input $x$ will exit at $i$th branch if the entropy of $y_i$ is larger than $t_i$. To choose $\{t_i\}_{i=1}^{K}$, Huang et al. (2018) provides a good starting point by fixing exiting probability of each branch classifiers equally on validation set so that each sample can equally contribute to inference. We follow this strategy but adjust the thresholds to make the contribution of middle branches slightly larger than the early branches. The threshold numbers for RDI-SmallCNN, RDI-ResNet38, and RDI-MobilenetV2 are set to be $\{0.023, 0.014\}$, $\{0.32, 0.36, 0.39, 0.83, 1.12, 1.35\}$, and $\{0.267, 0.765\}$, respectively.

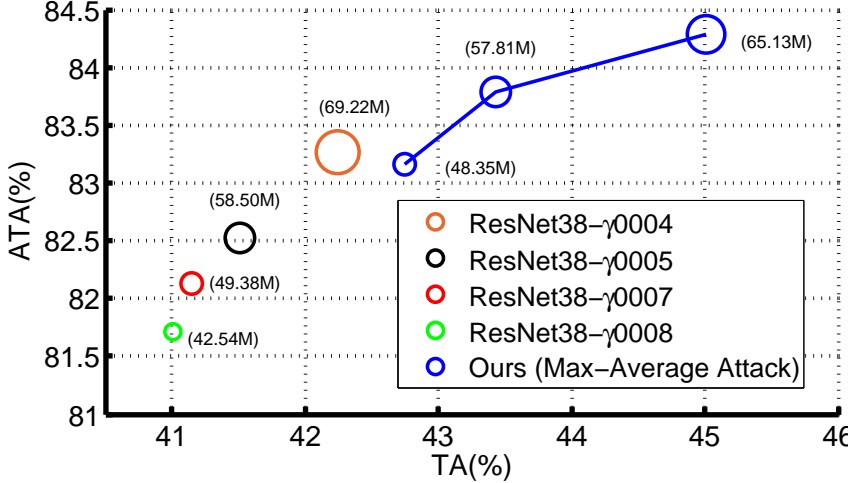

Figure 6: Performance comparison between RDI-Net and the pruning + defense baseline. Each marker represents a model, whose size is proportional to its MFlops. $\gamma$ is the sparsity trade-off parameter: the larger the sparser (smaller model).

## D    GENERALIZED ROBUSTNESS

Here, we introduce the attack form of random attack and report the complete results against FGSM (Goodfellow et al., 2015) and WRM (Sinha et al., 2018) attacker under various attack forms, in Tables 9 and 10, respectively.

**Random Attack**    the attack exploits multi-loss flexibility by randomly fusing all $f_{\theta_i}$ losses. Given a $N$-output network, we have a fusion vector $C \in \mathbb{R}^N \sim \mathbb{D}$, where $\mathbb{D}$ is some distribution (uniform

by default). We denote $c_j$ as the $j$th element of $C$ and $x_{rdm}^{adv}$ can be found by:

$$x_{rdm}^{adv} = \arg\max_{x' \in |x'-x|_\infty \leq \epsilon} |\frac{1}{N} \sum_{j=1}^{N} c_j \phi(f_{\theta_j}(x'), y)|. \tag{7}$$

It is expected to challenge our defense, due to the infinitely many ways of randomly fusing outputs.

Table 9: The performance evaluation on RDI-ResNet38 (defended with PGD) against FGSM attack. The perturbation size is $8/255$. The ATA of the original defended ResNet38 by PGD under the same attacker is $51.11\%$.

| Defence Method | Standard | Main Branch | Average | Max-Average |
|---|---|---|---|---|
| **TA** | 92.43% | 83.74% | 82.42% | **83.79%** |
| ATA (Branch1) | 20.69% | 66.06% | 72.77% | 72.76% |
| ATA (Branch2) | 16.15% | 53.87% | 70.40% | 69.71% |
| ATA (Branch3) | 8.13% | 63.70% | 64.19% | 65.14% |
| ATA (Branch4) | 10.09% | 56.67% | 58.45% | 58.20% |
| ATA (Branch5) | 9.45% | 50.81% | 52.76% | 52.96% |
| ATA (Branch6) | 10.22% | 50.34% | 53.17% | 51.05% |
| ATA (Main Branch) | 11.51% | 51.45% | 53.64% | 54.72% |
| ATA (Average) | 11.41% | 50.21% | 51.81% | 53.20% |
| ATA (Max-Average) | 2.09% | 47.53% | 50.63% | 52.40% |
| **ATA (Worst-Case)** | 2.09% | 47.53% | 50.63% | **51.05%** |
| Average MFlops | 65.74 | 55.27 | 58.27 | 59.67 |
| **Computation Saving** | 17.21% | 30.40% | 26.40% | 24.86% |

Table 10: The performance evaluation on RDI-ResNet38 (defended with PGD) against WRM attack. The perturbation size is $0.3$. The ATA of the original defended ResNet38 by PGD under the same attacker is $83.35\%$.

| Defence Method | Standard | Main Branch | Average | Max-Average |
|---|---|---|---|---|
| **TA** | 92.43% | 83.74% | 82.42% | **83.79%** |
| ATA (Branch1) | 46.60% | 83.73% | 82.42% | 83.78% |
| ATA (Branch2) | 71.33% | 83.73% | 82.42% | 83.79% |
| ATA (Branch3) | 23.51% | 83.73% | 82.41% | 83.78% |
| ATA (Branch4) | 33.41% | 83.73% | 82.42% | 83.78% |
| ATA (Branch5) | 42.35% | 83.73% | 82.41% | 83.78% |
| ATA (Branch6) | 47.77% | 83.74% | 82.40% | 83.78% |
| ATA (Main Branch) | 34.42% | 83.74% | 82.42% | 83.78% |
| ATA (Average) | 26.48% | 83.69% | 82.36% | 83.77% |
| ATA (Max-Average) | 23.51% | 83.73% | 82.40% | 83.78% |
| **ATA (Worst-Case)** | 23.51% | 83.69% | 82.36% | **83.77%** |
| Average MFlops | 50.05 | 50.46 | 52.89 | 52.38 |
| **Computation Saving** | 36.98% | 36.46% | 33.40% | 34.04% |

