# OpenReview forum: "Triple Wins: Boosting Accuracy, Robustness and Efficiency Together by Enabling Input-Adaptive Inference"
_ICLR.cc/2020/Conference — Accept (Poster)_

### Official Review · AnonReviewer1 · 2019-10-23
**Official Blind Review #1**

**Rating:** 8

**Review:**

The paper exploited input-adaptive multiple early-exits, an idea drawn from the efficient CNN inference, to the field of adversarial attack and defense. It is well-motivated by the dilemma between the large model capacity required by accurate and robust classification, and the resulting model complexity as well as inference latency.

Overall, this paper presents an interesting perspective, with strong results. The usage of input-adaptive inference reduces the average inference complexity, without conflicting the "larger capacity" assumption for co-winning robustness and accuracy.

Since no literature has discussed the attacks for a multi-exit network, the authors constructed three attack forms, and then utilized adversarial training to defend correspondingly. The design of Max-Average Attack is particularly smart - to balance between "benefiting all" and "maximally boosting one" (its result is also convincingly good).

The authors presented three groups of experiments, from relatively heavy networks (ResNet38), to very compact ones (MobileNet-V2). It is especially meaningful to see their strategy work on MobileNet too (though the computational saving is a bit less, no surprise). The authors also did due diligence in ablation study and comparing with recent alternatives.

Several points that could be addressed to potentially improve the paper:

- The authors want to make it clearer that: their "triple win" is not about constructing a light-weight model that is both accurate and robust. It's instead about given an accurate + robust, yet heavy-weight model, how to reduce its AVERAGE computational load per sample inference, by routing "easier" examples to earlier exits.

- Can the authors think of and construct more diverse and stronger attacks for RDI-Nets? For example, it would be interesting to attacking RDI-Nets (e.g., defended by Max-Average) with randomized weighted combinations of single attacks?

Note that, at inference time, the same "randomized combination" cannot be also adopted as defense, because an input always wants to exit the earliest possible for efficiency gains.

- The advantage over ATMC is not obvious: slightly lower TA, slightly higher ATA, and slightly more parameters. Could the authors try to align their parameters more closely (to the extent possible)?

- A missing related work: "Shallow-Deep Networks: Understanding and Mitigating Network Overthinking", ICML 2019. It also discussed how to append early exits to pre-trained backbones.




**Experience Assessment:**

I have published one or two papers in this area.

**Review Assessment: Checking Correctness Of Derivations And Theory:**

I carefully checked the derivations and theory.

**Review Assessment: Checking Correctness Of Experiments:**

I carefully checked the experiments.

**Review Assessment: Thoroughness In Paper Reading:**

I read the paper thoroughly.

---

> ### Author Response · Authors · 2019-11-12
> **Response to Reviewer#1**
>
>
> Q1. About highlighting our contribution
>
> You are precisely correct.  We reduce the average computation loads by input-adaptive routing to achieve "triple wins", rather than proposing a specific design of robust light-weight models. We have highlighted the difference in the revised paper (Section 3 beginning).
>
> Q2: More diverse attack forms.
> We appreciate your insightful suggestion. As you suggested, we create the new "randomized attack": that attack will randomly combine the multi-exit losses, where the weight coefficients are i.i.d. sampled from Gaussians and then normalized to have their sum equal one. On our strongest defended model of RDI-Resnet38 (using Max-Average), it achieves TA = 83.79%, ATA = 44.86%, with 28.88% MFlops saving.
>
> We also tried a direct attack on the decision function and report the results in the response to Reviewer #3.
>
> Q3: Closer comparison with ATMC
> We communicated with the ATMC authors and obtained their original implementation, to train a new model whose number of flops is much closer (to the extent possible) to our RDI-ResNet38. The results below demonstrate that ours outperforms ATMC in this setting. We have confirmed the results with ATMC authors.
> ------------------------------------
> Model    ATA        TA                  MFlops
> ATMC     42.66%  83.51%        58.03
> Ours      43.32%   83.79%         57.81
> ------------------------------------
>
> Q4: Missing citation.
> We appreciate your suggestion and have cited it.

---

### Official Review · AnonReviewer2 · 2019-10-24
**Official Blind Review #2**

**Rating:** 8

**Review:**

This paper considers the following problem: in a classification setting, it appears that by increasing the model capacity, the model accuracy and robustness seem to be improved, at the expense of model size and latency. Thus, the authors want to design an approach where at the same time accuracy, robustness and efficiency are improved at the same time.

Their idea is "multi-exit networks" with inference that adapts based on the input. Particularly, their proposed "Robust Dynamic Inference Networks" allows each input  -- clean or adversarial -- to choose adaptively one of the multiple output layers to output its prediction. This way, they can do an investigation to new variations of adversarial attacks and adversarial defenses. Their experiments show that indeed via this approach, they can achieve the triple wins of accuracy, robustness, and efficiency.

+ novel idea, promising results,
- Although I like the discussion of accuracy-robustness tradeoff in par 2 of Introduction, I am not sure about the statement that increasing model capacity both robustness and accuracy are improved, as used in the abstract, is always true.
+ First time adversarial attacks and defenses are studied in a multi-output model.
+ Interesting connection of multi-output networks with ensemble models.

Overall, I believe that this is an interesting, novel paper, which could be of high interest in the ICLR community, and I would vote for its acceptance.

**Experience Assessment:**

I have published one or two papers in this area.

**Review Assessment: Checking Correctness Of Derivations And Theory:**

I assessed the sensibility of the derivations and theory.

**Review Assessment: Checking Correctness Of Experiments:**

I assessed the sensibility of the experiments.

**Review Assessment: Thoroughness In Paper Reading:**

I read the paper at least twice and used my best judgement in assessing the paper.

---

> ### Author Response · Authors · 2019-11-12
> **Response to Reviewer#2**
>
>
> We sincerely appreciate your positive opinion and insightful suggestions about our work.
>
> Regarding the role of "increasing capacity", we draw the conclusions from the theoretical analysis in (Tsipras et al. (2019), Nakkiran (2019), which are aligned with our current observations in experiments. That said, those works present more a high-level motivation than any actual algorithmic foundation, for our work.

---

### Official Review · AnonReviewer3 · 2019-11-04
**Official Blind Review #3**

**Rating:** 3

**Review:**

This paper proposes a framework coined as ‘Robust Dynamic Inference Networks (RDI-Nets)’. The goal is concurrently achieving accuracy, robustness and efficiency via ‘input-adaptive inference’ and ‘multi-loss flexibility’  on a multi-output architecture. The observation is that
in a deep architecture, the representations in earlier layers can also be used for solving a specific downstream classification task. So by attaching several final classification stages at the intermediate layers and by using the uncertainty of the softmax output as a decision criteria as when to use the current output as the final decision, the authors aim to achieve a triple win.
The paper then proposes some attack criteria for a multi-output network.

The paper is not very well written. The paper has a designated related work section but the entire paper reads from its abstract to conclusions constantly like a literature review. This makes it hard to focus and identify the original contribution. The proposed architecture is only introduced later in detail in 3.3 after the attacks. I found the organization and writing style not very reader friendly.

The authors provide a large experimental section, however the key problem with the paper is that it blurs the evaluation issue. While the observation of using uncertainty of estimates at intermediate levels has some intuitive appeal, the decision criteria that the authors propose requires careful selection of thresholds and a good calibration. But given the thresholds the final decision is just a function of the entire network - as it should be. So a natural attack here is just attacking this decision function (or an approximate differentiable proxy) to see if this model provides extra robustness. Is such an evaluation available? Otherwise the proposed approach provides a false sense of robustness as the proposed attacks are not geared towards the actual underlying model.

Minor:
The definition of entropy in (6) is missing a minus sign.

The notation f(\theta| x) for theta as parameters and x as input for a function is in conflict with probability notation of conditional probabilities.


**Experience Assessment:**

I have read many papers in this area.

**Review Assessment: Checking Correctness Of Derivations And Theory:**

N/A

**Review Assessment: Checking Correctness Of Experiments:**

I assessed the sensibility of the experiments.

**Review Assessment: Thoroughness In Paper Reading:**

I made a quick assessment of this paper.

---

> ### Author Response · Authors · 2019-11-12
> **Response to Reviewer#3**
>
>
> Q1:  Paper organization and writing style.
> Thank you very much for your suggestion. We have revised the paper in an effort to improve its clarity and reader friendliness:
>
> -	We have collected the two discussion paragraphs: "Intuition: Multi-Output Networks as Special Ensembles" and "Do Triple Wins Go Against the Model Capacity Needs?", into one dedicated section "Discussion and Analysis" after Section 4. We hope that resolves the current impression "the entire paper reads constantly like a literature review".
> -	We also re-organized Section 3, to first provide an overview of RDI-Nets, followed by discussing concrete attack and defense forms.
>
> We are more than happy to take any further suggestions to revise this manuscript.
>
> Q2:  Directly attack the decision function.
> We appreciate this insightful and important comment. We conduct the requested evaluation and see our strongest defense (Max-Average) still perform effectively under the new attack.
> The decision function of the multi-exit network for an input example is the single exit loss function through which that specific example is actually routed. In view of that, we implemented this "direct decision attack", by every time computing the adversarial perturbations w.r.t. the actual single exit. The resulting new attack thus can be viewed as an input-adaptive selection version of single attacks. On RDI-ResNet38 with Max-Average defense, the result is:
> ------------------------------------
> ATA        TA                MFlops
> 43.70%   83.31%        64.82
> ------------------------------------
> Clearly, this new form of attack is indeed considered more challenging by RDI-Nets, as more examples are now routed to higher-level exits for more scrutiny. It leads to the improved averaged inference cost (~10 MFlops higher than the max-average attack in Table 3), a model behavior aligned with our expectation.
> Despite so, for this new attack, the ATA of our RDI-Net is 1.41% higher than the original ResNet-38 (Table 1) and the TA is comparable (0.30% lower), with 18.3% computational savings on average. Thus it still achieves our aimed "triple wins", though understandably with fewer margins, under this new stronger attack.
> We would like to thank you again for bringing up this new attack possibility. We would be more than happy to include the above discussions, and possibly more results into our paper if you are in favor of so. We also tried another suggested randomized attack and report the results in the response to Reviewer #1.
>
> Q3:  About minor comments.
> We appreciate your careful proofreading and have revised the paper accordingly.

---

> > ### Comment · AnonReviewer3 · 2019-11-14
> > **Thank you for your careful response.**
> >
> > I have read the responses and I will consider the revised manuscript. I appreciate the reporting of the new results.

---

### Decision · Program_Chairs · 2019-12-19

**Decision:**

Accept (Poster)

**Comment:**

The authors develop a novel technique to train networks to be robust and accurate while still being efficient to train and evaluate. The authors propose "Robust Dynamic Inference Networks" that allows inputs to be adaptively routed to one of several output channels and thereby adjust the inference time used for any given input. They show

The line of investigation initiated by authors is very interesting and should open up a new set of research questions in the adversarial training literature.

The reviewers were in consensus on the quality of the paper and voted in favor of acceptance. One of the reviewers had concerns about the evaluation in the paper, in particular about whether carefully crafted attacks could break the networks studied by the authors. However, the authors performed additional experiments and revised the paper to address this concern to the satisfaction of the reviewer.

Overall, the paper contains interesting contributions and should be accepted.